# SELF-LABELLING VIA SIMULTANEOUS CLUSTERING AND REPRESENTATION LEARNING

**Yuki M. Asano**       **Christian Rupprecht**       **Andrea Vedaldi**

Visual Geometry Group
University of Oxford
`{yuki,chrisr,vedaldi}@robots.ox.ac.uk`

## ABSTRACT

Combining clustering and representation learning is one of the most promising approaches for unsupervised learning of deep neural networks. However, doing so naively leads to ill posed learning problems with degenerate solutions. In this paper, we propose a novel and principled learning formulation that addresses these issues. The method is obtained by maximizing the information between labels and input data indices. We show that this criterion extends standard cross-entropy minimization to an optimal transport problem, which we solve efficiently for millions of input images and thousands of labels using a fast variant of the Sinkhorn-Knopp algorithm. The resulting method is able to self-label visual data so as to train highly competitive image representations without manual labels. Our method achieves state of the art representation learning performance for AlexNet and ResNet-50 on SVHN, CIFAR-10, CIFAR-100 and ImageNet and yields the first self-supervised AlexNet that outperforms the supervised Pascal VOC detection baseline. Code and models are available[1].

## 1 INTRODUCTION

Learning from unlabelled data can dramatically reduce the cost of deploying machine learning algorithms to new applications, thus amplifying their impact in the real world. Self-supervision is an increasingly popular framework for learning without labels. The idea is to define pretext learning tasks that can be constructed from raw data alone, but that still result in neural networks that transfer well to useful applications.

Much of the research in self-supervision has focused on designing new pretext tasks. However, given supervised data such as ImageNet (Deng et al., 2009), the standard classification objective of minimizing the cross-entropy loss still results in better or at least as good pre-training than any of such methods (for a given amount of data and for a given model complexity). This suggests that the task of classification is sufficient for pre-training networks, provided that suitable data labels are available. In this paper, we thus focus on the problem of obtaining the labels automatically by designing a self-labelling algorithm.

Learning a deep neural network together while discovering the data labels can be viewed as *simultaneous clustering and representation learning*. The latter can be approached by combining cross-entropy minimization with an off-the-shelf clustering algorithm such as $K$-means. This is precisely the approach adopted by the recent DeepCluster method (Caron et al., 2018), which achieves excellent results in unsupervised representation learning. However, combining representation learning, which is a discriminative task, with clustering is not at all trivial. In particular, we show that the combination of cross-entropy minimization and $K$-means as adopted by DeepCluster cannot be described as the optimization of an overall learning objective; instead, there exist degenerate solutions that the algorithm avoids via particular implementation choices.

In order to address this technical shortcoming, in this paper, we contribute a new principled formulation for simultaneous clustering and representation learning. The starting point is to minimize a single loss, the cross-entropy loss, for learning the deep network *and* for estimating the data labels. This

---

[1]`https://github.com/yukimasano/self-label`

is often done in semi-supervised learning and multiple instance learning. However, when applied naively to the unsupervised case, it immediately leads to a degenerate solution where all data points are mapped to the same cluster.

We solve this issue by adding the constraint that the labels must induce an equipartition of the data, which we show maximizes the information between data indices and labels. We also show that the resulting label assignment problem is the same as optimal transport, and can therefore be solved in polynomial time by linear programming. However, since we want to scale the algorithm to millions of data points and thousands of labels, standard transport solvers are inadequate. Thus, we also propose to use a fast version of the Sinkhorn-Knopp algorithm for finding an approximate solution to the transport problem efficiently at scale, using fast matrix-vector algebra.

Compared to methods such as DeepCluster, the new formulation is more principled and allows to more easily demonstrate properties of the method such as convergence. Most importantly, via extensive experimentation, we show that our new approach leads to significantly superior results than DeepCluster, achieving the new state of the art for representation learning approaches. In fact, the method's performance surpasses others that use a single type of supervisory signal for self-supervision, and is on par or better than very recent contributions as well (Tian et al., 2019; He et al., 2019; Misra & van der Maaten, 2019; Oord et al., 2018).

## 2 RELATED WORK

Our paper relates to two broad areas of research: (a) self-supervised representation learning, and (b) more specifically, training a deep neural network using pseudo-labels, i.e. the assignment of a label to each image. We discuss closely related works for each.

**Self-supervised learning:** A wide variety of methods that do not require manual annotations have been proposed for the self-training of deep convolutional neural networks. These methods use various cues and proxy tasks namely, in-painting (Pathak et al., 2016), patch context and jigsaw puzzles (Doersch et al., 2015; Noroozi & Favaro, 2016; Noroozi et al., 2018; Mundhenk et al., 2017), clustering (Caron et al., 2018; Huang et al., 2019; Zhuang et al., 2019; Bautista et al., 2016), noise-as-targets (Bojanowski & Joulin, 2017), colorization (Zhang et al., 2016; Larsson et al., 2017), generation (Jenni & Favaro, 2018; Ren & Lee, 2018; Donahue et al., 2017; Donahue & Simonyan, 2019), geometry (Dosovitskiy et al., 2016), predicting transformations (Gidaris et al., 2018; Zhang et al., 2019) and counting (Noroozi et al., 2017). Most recently, contrastive methods have shown great performance gains, (Oord et al., 2018; Hénaff et al., 2019; Tian et al., 2019; He et al., 2019) by leveraging augmentation and adequate losses. In (Feng et al., 2019), predicting rotation (Gidaris et al., 2018) is combined with instance retrieval (Wu et al., 2018) and multiple tasks are combined in (Doersch & Zisserman, 2017).

**Pseudo-labels for images:** In the self-supervised domain, we find a spectrum of methods that either give each data point a unique label (Wu et al., 2018; Dosovitskiy et al., 2016) or train on a flexible number of labels with $K$-means (Caron et al., 2018), with mutual information (Ji et al., 2018) or with noise (Bojanowski & Joulin, 2017). In (Noroozi et al., 2018) a large network is trained with a pretext task and a smaller network is trained via knowledge transfer of the clustered data. Finally, (Bach & Harchaoui, 2008; Vo et al., 2019) use convex relaxations to regularized affine-transformation invariant linear clustering, but can not scale to larger datasets.

Our contribution is a simple method that combines a novel pseudo-label extraction procedure from raw data alone and the training of a deep neural network using a standard cross-entropy loss.

## 3 METHOD

We will first derive our self-labelling method, then interpret the method as optimizing labels and targets of a cross-entropy loss and finally analyze similarities and differences with other clustering-based methods.

### 3.1 SELF-LABELLING

Neural network pre-training is often achieved via a supervised data classification task. Formally, consider a deep neural network $x = \Phi(I)$ mapping data $I$ (e.g. images) to feature vectors $x \in \mathbb{R}^D$. The model is trained using a dataset (e.g. ImageNet) of $N$ data points $I_1, \ldots, I_N$ with corresponding

labels $y_1, \ldots, y_N \in \{1, \ldots, K\}$, drawn from a space of $K$ possible labels. The representation is followed by a *classification head* $h : \mathbb{R}^D \to \mathbb{R}^K$, usually consisting of a single linear layer, converting the feature vector into a vector of class scores. The class scores are mapped to class probabilities via the softmax operator:

$$p(y = \cdot | \boldsymbol{x}_i) = \text{softmax}(h \circ \Phi(\boldsymbol{x}_i)).$$

The model and head parameters are learned by minimizing the average cross-entropy loss

$$E(p | y_1, \ldots, y_N) = -\frac{1}{N} \sum_{i=1}^{N} \log p(y_i | \boldsymbol{x}_i). \tag{1}$$

Training with objective (1) requires a labelled dataset. When labels are unavailable, we require a *self-labelling* mechanism to assign the labels automatically.

In semi-supervised learning, self-labelling is often achieved by jointly optimizing (1) with respect to the model $h \circ \Phi$ and the labels $y_1, \ldots, y_N$. This can work if at least part of the labels are known, thus constraining the optimization. However, in the fully unsupervised case, it leads to a degenerate solution: eq. (1) is trivially minimized by assigning all data points to a single (arbitrary) label.

To address this issue, we first rewrite eq. (1) by encoding the labels as posterior distributions $q(y | \boldsymbol{x}_i)$:

$$E(p, q) = -\frac{1}{N} \sum_{i=1}^{N} \sum_{y=1}^{K} q(y | \boldsymbol{x}_i) \log p(y | \boldsymbol{x}_i). \tag{2}$$

If we set the posterior distributions $q(y | \boldsymbol{x}_i) = \delta(y - y_i)$ to be deterministic, the formulations in eqs. (1) and (2) are equivalent, in the sense that $E(p, q) = E(p | y_1, \ldots, y_N)$. In this case, optimizing $q$ is the same as reassigning the labels, which leads to the degeneracy. To avoid this, we add the constraint that the label assignments must partition the data in equally-sized subsets. Formally, the learning objective objective[2] is thus:

$$\min_{p,q} E(p, q) \quad \text{subject to} \quad \forall y : q(y | \boldsymbol{x}_i) \in \{0, 1\} \text{ and } \sum_{i=1}^{N} q(y | \boldsymbol{x}_i) = \frac{N}{K}. \tag{3}$$

The constraints mean that each data point $\boldsymbol{x}_i$ is assigned to exactly one label and that, overall, the $N$ data points are split uniformly among the $K$ classes.

The objective in eq. (3) is combinatorial in $q$ and thus may appear very difficult to optimize. However, this is an instance of the *optimal transport problem*, which can be solved relatively efficiently. In order to see this more clearly, let $P_{yi} = p(y | \boldsymbol{x}_i) \frac{1}{N}$ be the $K \times N$ matrix of joint probabilities estimated by the model. Likewise, let $Q_{yi} = q(y | \boldsymbol{x}_i) \frac{1}{N}$ be $K \times N$ matrix of assigned joint probabilities. Using the notation of (Cuturi, 2013), we relax matrix $Q$ to be an element of the *transportation polytope*

$$U(r, c) := \{Q \in \mathbb{R}_+^{K \times N} \mid Q \mathbb{1} = r, \ Q^\top \mathbb{1} = c\}. \tag{4}$$

Here $\mathbb{1}$ are vectors of all ones of the appropriate dimensions, so that $r$ and $c$ are the marginal projections of matrix $Q$ onto its rows and columns, respectively. In our case, we require $Q$ to be a matrix of conditional probability distributions that split the data uniformly, which is captured by:

$$r = \frac{1}{K} \cdot \mathbb{1}, \quad c = \frac{1}{N} \cdot \mathbb{1}.$$

With this notation, we can rewrite the objective function in eq. (3), up to a constant shift, as

$$E(p, q) + \log N = \langle Q, -\log P \rangle, \tag{5}$$

where $\langle \cdot \rangle$ is the Frobenius dot-product between two matrices and $\log$ is applied element-wise. Hence optimizing eq. (3) with respect to the assignments $Q$ is equivalent to solving the problem:

$$\min_{Q \in U(r,c)} \langle Q, -\log P \rangle. \tag{6}$$

---

[2]We assume for simplicity that $K$ divides $N$ exactly, but the formulation is easily extended to any $N \geq K$ by setting the constraints to either $\lfloor N/K \rfloor$ or $\lfloor N/K \rfloor + 1$, in order to assure that there is a feasible solution.

This is a *linear program*, and can thus be solved in polynomial time. Furthermore, solving this problem always leads to an integral solution despite having relaxed $Q$ to the continuous polytope $U(r, c)$, guaranteeing the exact equivalence to the original problem.

In practice, however, the resulting linear program is large, involving millions of data points and thousands of classes. Traditional algorithms to solve the transport problem scale badly to instances of this size. We address this issue by adopting a fast version (Cuturi, 2013) of the *Sinkhorn-Knopp algorithm*. This amounts to introducing a regularization term

$$\min_{Q \in U(r,c)} \langle Q, -\log P \rangle + \frac{1}{\lambda} \text{KL}(Q \| rc^\top), \tag{7}$$

where KL is the Kullback-Leibler divergence and $rc^\top$ can be interpreted as a $K \times N$ probability matrix. The advantage of this regularization term is that the minimizer of eq. (7) can be written as:

$$Q = \text{diag}(\alpha) P^\lambda \text{diag}(\beta) \tag{8}$$

where exponentiation is meant element-wise and $\alpha$ and $\beta$ are two vectors of scaling coefficients chosen so that the resulting matrix $Q$ is also a probability matrix (see (Cuturi, 2013) for a derivation). The vectors $\alpha$ and $\beta$ can be obtained, as shown below, via a simple matrix scaling iteration.

For very large $\lambda$, optimizing eq. (7) is of course equivalent to optimizing eq. (6), but even for moderate values of $\lambda$ the two objectives tend to have approximately the same optimizer (Cuturi, 2013). Choosing $\lambda$ trades off convergence speed with closeness to the original transport problem. In our case, using a fixed $\lambda$ is appropriate as we are ultimately interested in the final clustering and representation learning results, rather than in solving the transport problem exactly.

Our final algorithm's core can be described as follows. We learn a model $h \circ \Phi$ and a label assignment matrix $Q$ by solving the optimization problem eq. (6) with respect to both $Q$, which is a probability matrix, and the model $h \circ \Phi$, which determines the predictions $P_{yi} = \text{softmax}_y(h \circ \Phi(\boldsymbol{x}_i))$. We do so by alternating the following two steps:

**Step 1: representation learning.** Given the current label assignments $Q$, the model is updated by minimizing eq. (6) with respect to (the parameters of) $h \circ \Phi$. This is the same as training the model using the common cross-entropy loss for classification.

**Step 2: self-labelling.** Given the current model $h \circ \Phi$, we compute the log probabilities $P$. Then, we find $Q$ using eq. (8) by iterating the updates (Cuturi, 2013)

$$\forall y : \alpha_y \leftarrow [P^\lambda \beta]_y^{-1} \qquad \forall i : \beta_i \leftarrow [\alpha^\top P^\lambda]_i^{-1}.$$

Each update involves a single matrix-vector multiplication with complexity $\mathcal{O}(NK)$, so it is relatively quick even for millions of data points and thousands of labels and so the cost of this method scales linearly with the number of images $N$. In practice, convergence is reached within 2 minutes on ImageNet when computed on a GPU. Also, note that the parameters $\alpha$ and $\beta$ can be retained between steps, thus allowing a warm start of Step 2.

## 3.2 INTERPRETATION

As shown above, the formulation in eq. (2) uses scaled versions of the probabilities. We can interpret these by treating the data *index* $i$ as a random variable with uniform distribution $p(i) = 1/N$ and by rewriting the posteriors $p(y|\boldsymbol{x}_i) = p(y|i)$ and $q(y|\boldsymbol{x}_i) = q(y|i)$ as conditional distributions with respect to the data index $i$ instead of the feature vector $\boldsymbol{x}_i$. With these changes, we can rewrite eq. (5) as

$$E(p, q) + \log N = -\sum_{i=1}^{N} \sum_{y=1}^{K} q(y, i) \log p(y, i) = H(q, p), \tag{9}$$

which is the cross-entropy between the joint label-index distributions $q(y, i)$ and $p(y, i)$. The minimum of this quantity w.r.t. $q$ is obtained when $p = q$, in which case $E(q, q) + \log N$ reduces to the entropy $H_q(y, i)$ of the random variables $y$ and $i$. Additionally, since we assumed that $q(i) = 1/N$, the marginal entropy $H_q(i) = \log N$ is constant and, due to the equipartition condition $q(y) = 1/K$, $H_q(y) = \log K$ is *also* constant. Subtracting these two constants from the entropy yields:

$$\min_p E(p, q) + \log N = E(q, q) + \log N = H_q(y, i) = H_q(y) + H_q(i) - I_q(y, i) = \text{const.} - I_q(y, i).$$

Thus we see that minimizing $E(p, q)$ is the same as maximizing the *mutual information* between the label $y$ and the data index $i$.

In our formulation, the maximization above is carried out under the equipartition constraint. We can instead relax this constraint and directly maximize the information $I(y, i)$. However, by rewriting information as the difference $I(y, i) = H(y) - H(y|i)$, we see that the optimal solution is given by $H(y|i) = 0$, which states each data point $i$ is associated to only one label deterministically, and by $H(y) = \ln K$, which is another way of stating the equipartition condition.

In other words, our learning formulation can be interpreted as maximizing the information between data indices and labels while explicitly enforcing the equipartition condition, which is implied by maximizing the information in any case. Compared to minimizing the entropy alone, maximizing information avoids degenerate solutions as the latter carry no mutual information between labels $y$ and indices $i$. Similar considerations can be found in (Ji et al., 2018).

### 3.3 RELATION TO SIMULTANEOUS REPRESENTATION LEARNING AND CLUSTERING

In the discussion above, self-labelling amounts to assigning discrete labels to data and can thus be interpreted as clustering. Most of the traditional clustering approaches are *generative*. For example, $K$-means takes a dataset $\boldsymbol{x}_1, \ldots, \boldsymbol{x}_N$ of vectors and partitions it into $K$ classes in order to minimize the reconstruction error

$$E(\boldsymbol{\mu}_1, \ldots, \boldsymbol{\mu}_K, y_1, \ldots, y_N) = \frac{1}{N} \sum_{i=1}^{N} \|\boldsymbol{x}_i - \boldsymbol{\mu}_{y_i}\|^2 \qquad (10)$$

where $y_i \in \{1, \ldots, K\}$ are the data-to-cluster assignments and $\boldsymbol{\mu}_y$ are means approximating the vectors in the corresponding clusters. The $K$-means energy can thus be interpreted as the average data reconstruction error.

It is natural to ask whether a clustering method such as $K$-means, which is based on approximating the input data, could be combined with representation learning, which uses a discriminative objective. In this setting, the feature vectors $\boldsymbol{x} = \Phi(I)$ are extracted by the neural network $\Phi$ from the input data $I$. Unfortunately, optimizing a loss such as eq. (10) with respect to the clustering and representation parameters is meaningless: in fact, the obvious solution is to let the representation send all the data points to the same constant feature vector and setting all the means to coincide with it, in which case the $K$-means reconstruction error is zero (and thus minimal).

Nevertheless, DeepCluster (Caron et al., 2018) *does* successfully combine $K$-means with representation learning. DeepCluster can be related to our approach as follows. Step 1 of the algorithm, namely representation learning via cross-entropy minimization, is exactly the same. Step 2, namely self-labelling, differs: where we solve an optimal transport problem to obtain the pseudo-labels, they do so by running $K$-means on the feature vectors extracted by the neural network.

DeepCluster does have an obvious degenerate solution: we can assign all data points to the same label and learn a constant representation, achieving simultaneously a minimum of the cross-entropy loss in Step 1 and of the $K$-means loss in Step 2. The reason why DeepCluster avoids this pitfall is due to the particular interaction between the two steps. First, during Step 2, the features $\boldsymbol{x}_i$ are fixed so $K$-means cannot pull them together. Instead, the means spread to cover the features as they are, resulting in a balanced partitioning. Second, during the classification step, the cluster assignments $y_i$ are fixed, and optimizing the features $\boldsymbol{x}_i$ with respect to the cross-entropy loss tends to separate them. Lastly, the method in (Caron et al., 2018) also uses other heuristics such as sampling the training data inversely to their associated clusters' size, leading to further regularization.

However, a downside of DeepCluster is that it does not have a single, well-defined objective to optimize, which means that it is difficult to characterize its convergence properties. By contrast, in our formulation, both Step 1 and Step 2 optimize *the same objective*, with the advantage that convergence to a (local) optimum is guaranteed.

### 3.4 AUGMENTING SELF-LABELLING VIA DATA TRANSFORMATIONS

Methods such as DeepCluster extend the training data via augmentations. In vision problems, this amounts to (heavily) distorting and cropping the input images at random. Augmentations are applied so that the neural network is encouraged to learn a labelling function which is *transformation invariant*.

In practice, this is crucial to learn good clusters and representations, so we adopt it here. This is achieved by setting $P_{yi} = \mathbb{E}_t[\log \mathrm{softmax}_y h \circ \Phi(t\boldsymbol{x}_i)]$ where the transformations $t$ are sampled at random. In practice, in Step 1 (representation learning), this is implemented via the application of the random transformations to data batches during optimization via SGD, which is corresponds to the usual data augmentation scheme for deep neural networks. As noted in (YM. et al., 2020), and as can be noted by an analysis of recent publications (Hénaff et al., 2019; Tian et al., 2019; Misra & van der Maaten, 2019), augmentation is critical for good performance.

### 3.5 MULTIPLE SIMULTANEOUS SELF-LABELINGS

Intuitively, the same data can often be clustered in many equally good ways. For example, visual objects can be clustered by color, size, typology, viewpoint, and many other attributes. Since our main objective is to use clustering to learn a good data representation $\Phi$, we consider a multi-task setting in which the same representation is shared among several different clustering tasks, which can potentially capture different and complementary clustering axis.

In our formulation, this is easily achieved by considering *multiple heads* (Ji et al., 2018) $h_1, \ldots, h_T$, one for each of $T$ clustering tasks (which may also have a different number of labels). Then, we optimize a sum of objective functions of the type eq. (6), one for each task, while sharing the parameters of the feature extractor $\Phi$ among them.

## 4 EXPERIMENTS

In this section, we evaluate the quality of the representations learned by our *Self Labelling* (SeLa) technique. We first test variants of our method, including ablating its components, in order to find an optimal configuration. Then, we compare our results to the state of the art in self-supervised representation learning, where we find that our method is the best among clustering-based techniques and overall state-of-the-art or at least highly competitive in many benchmarks. In the appendix, we also show qualitatively that the labels identified by our algorithm are usually meaningful and group visually similar concepts in the same clusters, often even capturing whole ImageNet classes.

### 4.1 SETUP

**Linear probes.** In order to quantify if a neural network has learned useful feature representations, we follow the standard approach of using *linear probes* (Zhang et al., 2017). This amounts to solving a difficult task, such as ImageNet classification, by training a linear classifier on top of a pre-trained feature representation, which is kept fixed. Linear classifiers heavily rely on the quality of the representation since their discriminative power is low. We apply linear probes to all intermediate convolutional blocks of representative networks. While linear probes are conceptually straightforward, there are several technical details that can affect the final accuracy, so we follow the standard protocol further outlined in the Appendix.

**Data.** For training data we consider ImageNet LSVRC-12 (Deng et al., 2009) and other smaller scale datasets. We also test our features by transferring them to MIT Places (Zhou et al., 2014). All of these are standard benchmarks for evaluation in self-supervised learning.

**Architectures.** Our base encoder architecture is AlexNet (Krizhevsky et al., 2012), since this is the most frequently used in other self-supervised learning works for the purpose of benchmarking. We inject the probes right after the ReLU layer in each of the five blocks, and denote these entry points `conv1` to `conv5`. Furthermore, since the `conv1` and `conv2` can be learned effectively from data augmentations alone (YM. et al., 2020), we focus the analysis on the deeper layers `conv2` to `conv5` which are more sensitive to the quality of the learning algorithm. In addition to AlexNet, we also test ResNet-50 (He et al., 2016) models. Further experimental details are given in the Appendix.

### 4.2 OPTIMAL CONFIGURATION AND ABLATIONS

In tables 1 and 5, we first validate various modelling and configuration choices. Two key hyper-parameters are the number of clusters $K$ and the number of clustering heads $T$, which we denote in the experiments below with the shorthand "SeLa[$K \times T$]". We run SeLa by alternating steps 1 and 2 as described in section 3.1. Step 1 amounts to standard CE training, which we run for a fixed number of epochs. Step 2 can be interleaved at any point in the optimization; to amortize its cost, we run it

Table 1: **Ablation: number of self-labelling steps.**

| Method | #opt. | c3 | c4 | c5 |
|---|---|---|---|---|
| SeLa [3k × 1] | 0 | 20.8 | 18.3 | 13.4 |
| SeLa [3k × 1] | 40 | 42.7 | 43.4 | 39.2 |
| SeLa [3k × 1] | 80 | 43.0 | 44.7 | 40.9 |
| SeLa [3k × 1] | 160 | 42.4 | 44.6 | 40.7 |

Table 2: **Number of clusters $K$.**

| Method | c3 | c4 | c5 |
|---|---|---|---|
| SeLa [1k × 1] | 40.1 | 42.1 | 38.8 |
| SeLa [3k × 1] | 43.0 | 44.7 | 40.9 |
| SeLa [5k × 1] | 42.5 | 43.9 | 40.2 |
| SeLa [10k × 1] | 42.2 | 43.8 | 39.7 |

Table 3: **Ablation: number of heads $T$.** (`c4` for AlexNet)

| Method | Architecture | Top-1 |
|---|---|---|
| SeLa [3k × 1] | AlexNet | 44.7 |
| SeLa [3k × 10] | AlexNet | 46.7 |
| SeLa [3k × 1] | ResNet-50 | 51.8 |
| SeLa [3k × 10] | ResNet-50 | 61.5 |

Table 4: **Different architectures.**

| Method | Architecture | Top-1 |
|---|---|---|
| SeLa [3k × 1] | AlexNet (small) | 41.3 |
| SeLa [3k × 1] | AlexNet | 44.7 |
| SeLa [3k × 1] | ResNet-50 | 51.8 |

Table 5: **Label transfer.**

| Method | Source (Top-1) | Target (Top-1) |
|---|---|---|
| SeLa [3k × 10] | AlexNet (46.7) | AlexNet (46.5) |
| SeLa [3k × 1] | ResNet-50 (51.8) | AlexNet (45.0) |
| SeLa [3k × 10] | ResNet-50 (61.5) | AlexNet (48.4) |

at most once per epoch, and usually less, with a schedule described and validated below. For these experiments, we train the representation and the linear probes on ImageNet.

**Number of clusters $K$.** Table 2, compares different values for $K$: moving from 1k to 3k improves the results, but larger numbers decrease the quality slightly.

**Ablation: number of heads $T$.** Table 3 shows that increasing the number of heads from $T = 1$ to $T = 10$ yields a large performance gain: $+2\%$ for AlexNet and $+10\%$ for ResNet. The latter more expressive model appears to benefit more from a more diverse training signal.

**Ablation: number of self-labelling iterations.** First, in table 1, we show that self-labelling (step 2) is essential for good performance, as opposed to only relying on the initial random label assignments and the data augmentations. For this, we vary the number of times the self-labelling algorithm (step 2) is run during training (#opts), from zero to once per step 1 epoch. We see that self-labelling is essential, with the best value around 80 (for 160 step 1 epochs in total).

**Architectures.** Table 4 compares a smaller variant of AlexNet which uses (64, 192) filters in its first two convolutional layers (Krizhevsky, 2014), to the standard variant with (96, 256) (Krizhevsky et al., 2012), all the way to a ResNet-50. SeLa works well in all cases, for large models such as ResNet but also smaller ones such as AlexNet, for which methods such as BigBiGAN (Donahue & Simonyan, 2019) or CPC (Hénaff et al., 2019) are unsuitable.

## 4.3 LABEL TRANSFER

An appealing property of SeLa is that the label it assigns to the images can be used to train another model from scratch, using standard supervised training. For instance, table 5 shows that, given the labels assigned by applying SeLa to AlexNet, we can re-train AlexNet from scratch using a shorter 90-epochs schedule with achieving the same final accuracy. This shows that the quality of the learned representation depends only the final label assignment, not on the fact that the representation is learned jointly with the labels. More interestingly, we can transfer labels between different architectures. For example, the labels obtained by applying SeLa [3k × 1] and SeLa [3k × 10] to ResNet-50 can be used to train a better AlexNet model than applying SeLa to the latter directly. For this reason, we publish on our website the self-labels for the ImageNet dataset in addition to the code and trained models.

## 4.4 SMALL-SCALE DATASETS

Here, we evaluate our method on relatively simple and small datasets, namely CIFAR-10/100 (Krizhevsky et al., 2009) and SVHN (Netzer et al., 2011). For this, we follow the experimental and evaluation protocol from the current state of the art in self-supervised learning in these datasets, AND (Huang et al., 2019). In table 6, we compare our method with the settings [128 × 10] for CIFAR-10, [512 × 10] for CIFAR-100 and [128 × 1] for SVHN to other published methods; details on the evaluation method are provided in the appendix. We observe that our proposed method outperforms the best previous method by $5.8\%$ for CIFAR-10, by $9.5\%$ for CIFAR-100 and by $0.8\%$ for SVHN when training a linear classifier on top of the frozen network. The relatively minor gains on SVHN can be explained by the fact that the gap between the supervised

Table 6: Nearest Neighbour and linear classification evaluation on small datasets using AlexNet. Results of previous methods are taken from (Huang et al., 2019).

|  | **Dataset** | | |
|---|---|---|---|
| **Method** | CIFAR-10 | CIFAR-100 | SVHN |
| Classifier/Feature | Linear Classifier / conv5 | | |
| *Supervised* | 91.8 | 71.0 | 96.1 |
| Counting | 50.9 | 18.2 | 63.4 |
| DeepCluster | 77.9 | 41.9 | 92.0 |
| Instance | 70.1 | 39.4 | 89.3 |
| AND | 77.6 | 47.9 | 93.7 |
| SL | **83.4** | **57.4** | **94.5** |
| Classifier/Feature | Weighted kNN / FC | | |
| *Supervised* | 91.9 | 69.7 | 96.5 |
| Counting | 41.7 | 15.9 | 43.4 |
| DeepCluster | 62.3 | 22.7 | 84.9 |
| Instance | 60.3 | 32.7 | 79.8 |
| AND | 74.8 | 41.5 | 90.9 |
| SL | **77.6** | **44.2** | **92.8** |

Table 7: **PASCAL VOC finetuning.** VOC07-Classification %mAP, VOC07-Detection %mAP and VOC12-Segmentation %mIU. * denotes a larger AlexNet variant.

|  | PASCAL VOC Task | | | |
|---|---|---|---|---|
| **Method** | **Cls.** | | **Det.** | **Seg.** |
|  | fc6-8 | all | all | all |
| ImageNet labels | 78.9 | 79.9 | 59.1 | 48.0 |
| Random | − | 53.3 | 43.4 | − |
| Random Rescaled | − | 56.6 | 45.6 | 32.6 |
| BiGAN | 52.3 | 60.1 | 46.9 | 35.2 |
| Context* | 55.1 | 65.3 | 51.1 | − |
| Context 2 | − | 69.6 | 55.8 | 41.4 |
| CC+VGG | − | 72.5 | 56.5 | 42.6 |
| RotNet | 70.9 | 73.0 | 54.4 | 39.1 |
| DeepCluster* | 72.0 | 73.4 | 55.4 | 45.1 |
| RotNet+retrieval* | 72.5 | 74.7 | 58.0 | **45.9** |
| SeLa* [3k × 10] | 73.1 | 75.3 | 55.9 | 43.7 |
| SeLa* [3k × 10]⁻ | 74.4 | 75.9 | 57.8 | 44.7 |
| SeLa* [3k × 10]⁻+Rot | **75.6** | **77.2** | **59.2** | 45.7 |

Table 8: Nearest Neighbour and linear classification evaluation using imbalanced CIFAR-10 training data. We evaluate on the normal CIFAR-10 test set and on CIFAR-100 to analyze the transferability of the features. Difference to the supervised baseline in parentheses. See section 4.5 for details.

|  | kNN | | Linear/conv5 | |
|---|---|---|---|---|
| **Training data/Method** | CIFAR-10 | CIFAR-100 | CIFAR-10 | CIFAR-100 |
| *CIFAR-10, full* | | | | |
| Supervised | 92.1 | 24.0 | 90.2 | 54.2 |
| ours ($K$-means) [128 × 1] | 64.7 $(-17.4)$ | 19.3 $(-4.7)$ | 77.5 $(-12.7)$ | 45.6 $(-8.8)$ |
| ours (SK) [128 × 1] | 72.9 $(-9.2)$ | 28.9 $(+4.9)$ | 79.8 $(-9.4)$ | 49.4 $(-4.8)$ |
| *CIFAR-10, light imbalance* | | | | |
| Supervised | 92.0 | 24.0 | 90.4 | 53.6 |
| ours ($K$-means) [128 × 1] | 64.2 $(-17.8)$ | 18.1 $(-5.9)$ | 77.0 $(-13.4)$ | 44.8 $(-8.8)$ |
| ours (SK) [128 × 1] | 71.7 $(-10.3)$ | 28.2 $(+4.2)$ | 79.5 $(-10.9)$ | 48.6 $(-5.0)$ |
| *CIFAR-10, heavy imbalance* | | | | |
| Supervised | 86.7 | 22.6 | 86.8 | 51.4 |
| ours ($K$-means) [128 × 1] | 60.7 $(-16.0)$ | 17.8 $(-4.8)$ | 75.2 $(-11.6)$ | 44.3 $(-7.1)$ |
| ours (SK) [128 × 1] | 67.6 $(-9.1)$ | 26.7 $(+3.9)$ | 77.2 $(-9.6)$ | 47.5 $(-2.9)$ |

baseline and the self-supervised results is already very small ($<3\%$). We also evaluate our method using weighted kNN using an embedding of size 128. We find that the proposed method consistently outperforms the previous state of the art by around $2\%$ across these datasets, even though AND is based on explicitly learning local neighbourhoods.

## 4.5 IMBALANCED DATA EXPERIMENTS

In order to understand if our equipartition regularization is affected by the underlying class distribution of a dataset, we perform multiple ablation experiments on artificially imbalanced datasets in table 8. We consider three training datasets based on CIFAR-10. The first is the original dataset with 5000 images for each class (*full* in table 8). Second, we remove $50\%$ of the images of one class (truck) while the rest remains untouched (*light imbalance*) and finally we remove $10\%$ of one class, $20\%$ of the second class and so on (*heavy imbalance*). On each of the three datasets we compare the performance of our method "ours (SK)" with a baseline that replaces our Sinkhorn-Knopp optimization with $K$-means clustering "ours ($K$-means)". We also compare the performance to training a network under full supervision. The evaluation follows Huang et al. (2019) and is based on linear probing and kNN classification — both on CIFAR-10 and, to understand feature generalization, on CIFAR-100.

Table 9: **Linear probing evaluation – AlexNet.** A linear classifier is trained on the (downsampled) activations of each layer in the pretrained model. We **bold** the best result in each layer and underline the second best. The best layer is highlighted in blue. * denotes a larger AlexNet variant. ⁻ refers to AlexNets trained with self-label transfer from a corresponding ResNet-50. "+Rot" refers to retraining using labels and an additional RotNet loss, "+ more aug." includes further augmentation during retraining. See Table A.2 in the Appendix for a full version of this table and details.

| | Method | ILSVRC-12 | | | | | Places | | | | |
|---|---|---|---|---|---|---|---|---|---|---|---|
| | | c1 | c2 | c3 | c4 | c5 | c1 | c2 | c3 | c4 | c5 |
| | ImageNet supervised, (Zhang et al., 2017) | 19.3 | 36.3 | 44.2 | 48.3 | 50.5 | 22.7 | 34.8 | 38.4 | 39.4 | 38.7 |
| | Places supervised, (Zhang et al., 2017) | - | - | - | - | - | 22.1 | 35.1 | 40.2 | 43.3 | 44.6 |
| | Random, (Zhang et al., 2017) | 11.6 | 17.1 | 16.9 | 16.3 | 14.1 | 15.7 | 20.3 | 19.8 | 19.1 | 17.5 |
| *1-crop evaluation* | Inpainting, (Pathak et al., 2016) | 14.1 | 20.7 | 21.0 | 19.8 | 15.5 | 18.2 | 23.2 | 23.4 | 21.9 | 18.4 |
| | BiGAN, (Donahue et al., 2017) | 17.7 | 24.5 | 31.0 | 29.9 | 28.0 | 22.0 | 28.7 | 31.8 | 31.3 | 29.7 |
| | Instance retrieval, (Wu et al., 2018) | 16.8 | 26.5 | 31.8 | 34.1 | 35.6 | 18.8 | 24.3 | 31.9 | 34.5 | 33.6 |
| | RotNet, (Gidaris et al., 2018) | 18.8 | 31.7 | 38.7 | 38.2 | 36.5 | 21.5 | 31.0 | 35.1 | 34.6 | 33.7 |
| | AND*,(Huang et al., 2019) | 15.6 | 27.0 | 35.9 | 39.7 | 37.9 | - | - | - | - | - |
| | CMC*,(Tian et al., 2019) | 18.4 | 33.5 | 38.1 | 40.4 | 42.6 | - | - | - | - | - |
| | AET*,(Zhang et al., 2019) | 19.3 | 35.4 | 44.0 | 43.6 | 42.4 | 22.1 | 32.9 | 37.1 | 36.2 | 34.7 |
| | RotNet+retrieval*, (Feng et al., 2019) | 20.8 | 35.2 | 41.8 | 44.3 | 44.4 | 24.0 | 33.8 | 37.5 | 39.3 | 38.9 |
| | SeLa [3k × 10]* | 20.3 | 32.2 | 38.6 | 41.4 | 39.6 | 24.5 | 31.9 | 36.7 | 38.0 | 37.0 |
| | SeLa [3k × 10]⁻+Rot* | 20.6 | 32.3 | 40.4 | 43.1 | 42.3 | 24.0 | 31.7 | 37.1 | 39.0 | 37.6 |
| | SeLa [3k × 10]⁻+Rot*+more aug. | 19.2 | 32.6 | 40.8 | 44.4 | 44.7 | 21.1 | 30.4 | 36.5 | 37.9 | 37.3 |
| *10-crop evaluation* | ImageNet supervised* | 21.6 | 37.2 | 46.9 | 52.9 | 54.4 | 22.6 | 33.2 | 39.0 | 41.3 | 39.7 |
| | DeepCluster*, (Caron et al., 2018) | 13.4 | 32.3 | 41.0 | 39.6 | 38.2 | 23.8 | 32.8 | 37.3 | 36.0 | 31.0 |
| | Local Agg.*, (Zhuang et al., 2019) | 18.7 | 32.7 | 38.1 | 42.3 | 42.4 | 18.7 | 32.7 | 38.2 | 40.3 | 39.5 |
| | RotNet+retrieval*, (Feng et al., 2019) | 22.2 | 38.2 | 45.7 | 48.7 | 48.3 | 25.5 | 36.0 | 40.1 | 42.2 | 41.3 |
| | SeLa [3k × 10]* | 22.5 | 37.4 | 44.7 | 47.1 | 44.1 | 26.7 | 34.9 | 39.9 | 41.8 | 39.7 |
| | SeLa [3k × 10]⁻+Rot* | 22.8 | 37.8 | 46.7 | 49.7 | 48.4 | 26.8 | 35.5 | 41.0 | 43.0 | 41.3 |
| | SeLa [3k × 10]⁻+Rot*+more aug. | 21.9 | 37.1 | 46.0 | 50.0 | 50.0 | 23.4 | 33.0 | 39.4 | 41.4 | 39.9 |

To our surprise, we find that our method generalizes better to CIFAR-100 than the supervised baseline during kNN evaluation, potentially due to overfitting when training with labels. We also find that using SK optmization for obtaining pseudo-labels is always better than $K$-means on all metrics and datasets. When comparing the imbalance settings, we find that under the *light imbalance* scenario, the methods' performances are ranked the same and no method is strongly affected by the imbalance. Under the *heavy imbalance* scenario, all methods drop in performance. However, compared to full data and light imbalance, the gap between supervised and self-supervised even decreases slightly for both $K$-means and our method, indicating stronger robustness of self-supervised methods compared to a supervised one.

In conclusion, our proposed method does not rely on the data to contain the same number of images for every class and outperforms a $K$-means baseline even in very strong imbalance settings. This confirms the intuition that the equipartitioning constraint acts as a regularizer and does not exploit the class distribution of the dataset.

## 4.6 LARGE SCALE BENCHMARKS

To compare to the state of the art and concurrent work, we evaluate several architectures using linear probes on public benchmark datasets.

**AlexNet.** The main benchmark for feature learning methods is linear probing of an AlexNet trained on ImageNet. In table 9 we compare the performance across layers also on the Places dataset. We find that across both datasets our method outperforms DeepCluster and local Aggregation at every layer. From our ablation studies in tables 1-5 we also note that even our single head variant [3k × 1] outperforms both methods. Given that our method provides labels for a dataset that can be used for retraining a network quickly, we find that we can improve upon this initial performance. And by adopting a hybrid approach, similar to (Feng et al., 2019), of training an AlexNet with 10 heads and one additional head for computing the RotNet loss, we find further improvement. This result (SeLa

Table 10: **Linear evaluation - ResNet.** A linear layer is trained on top of the global average pooled features of ResNets. All evaluations use a single centred crop. We have separated much larger architectures such as RevNet-50×4 and ResNet-161. Methods in brackets use a augmentation policy learned from supervised training and methods with * are not explicit about which further augmentations they use. See Table A.3 in the Appendix for a full version of this table.

| Method | Architecture | Top-1 | Top-5 |
|---|---|---|---|
| Supervised, (Donahue & Simonyan, 2019) | ResNet-50 | 76.3 | 93.1 |
| Jigsaw, (Kolesnikov et al., 2019) | ResNet-50 | 38.4 | − |
| Rotation, (Kolesnikov et al., 2019) | ResNet-50 | 43.8 | − |
| CPC, (Oord et al., 2018) | ResNet-101 | 48.7 | 73.6 |
| BigBiGAN, (Donahue & Simonyan, 2019) | ResNet-50 | 55.4 | 77.4 |
| LocalAggregation, (Zhuang et al., 2019) | ResNet-50 | 60.2 | − |
| Efficient CPC v2.1, (Hénaff et al., 2019) | ResNet-50 | (63.8) | (85.3) |
| CMC, (Tian et al., 2019) | ResNet-50 | (64.1) | (85.4) |
| MoCo, (He et al., 2019) | ResNet-50 | 60.6 | − |
| PIRL, (Misra & van der Maaten, 2019)* | ResNet-50 | **63.6** | − |
| SeLa [3k × 10] | ResNet-50 | 61.5 | **84.0** |
| *other architectures* | | | |
| MoCo, (He et al., 2019) | RevNet-50×4 | 68.6 | − |
| Efficient CPC v2.1, (Hénaff et al., 2019) | ResNet-161 | **71.5** | **90.1** |

[3k × 10]⁻+Rot) achieves state of the art in unsupervised representation learning for AlexNet, with a gap of $1.3\%$ to the previous best performance on ImageNet and surpasses the ImageNet supervised baseline transferred to Places by $1.7\%$.

**ResNet.** Training better models than AlexNets is not yet standardized in the feature learning community. In Table 10 we compare a ResNet-50 trained with our method to other works. With top-1 accuracy of $61.5$, we outperform than all other methods including Local Aggregation, CPCv1 and MoCo that use the same level of data augmentation. We even outperform larger architectures such as BigBiGAN's RevNet-50x4 and reach close to the performance of models using AutoAugment-style transformations.

## 4.7 Fine-tuning: Classification, object detection and semantic segmentation

Finally, since pre-training is usually aimed at improving down-stream tasks, we evaluate the quality of the learned features by fine-tuning the model for three distinct tasks on the PASCAL VOC benchmark. In Table 7 we compare results with regard to multi-label classification, object detection and semantic segmentation on PASCAL VOC (Everingham et al., 2010).

As in the linear probe experiments, we find our method better than the current state of the art in detection and classification with both fine-tuning only the last fully connected layers and when fine-tuning the whole network ("*all*"). Notably, our fine-tuned AlexNet outperforms its supervised ImageNet baseline on the VOC detection task. Also for segmentation the method is very close ($0.2\%$) to the best performing method. This shows that our trained network does not only learn useful feature representations but is also able to perform well when fine-tuned on actual down-stream tasks.

## 5 Conclusion

We present a self-supervised feature learning method that is based on clustering. In contrast to other methods, ours optimizes the same objective during feature learning and during clustering. This becomes possible through a weak assumption that the number of samples should be equal across clusters. This constraint is explicitly encoded in the label assignment step and can be solved for efficiently using a modified Sinkhorn-Knopp algorithm. Our method outperforms all other feature learning approaches and achieves SOTA on SVHN, CIFAR-10/100 and ImageNet for AlexNet and ResNet-50. By virtue of the method, the resulting self-labels can be used to quickly learn features for new architectures using simple cross-entropy training.

ACKNOWLEDGMENTS

Yuki Asano gratefully acknowledges support from the EPSRC Centre for Doctoral Training in Autonomous Intelligent Machines & Systems (EP/L015897/1). We are also grateful to ERC IDIU-638009, AWS Machine Learning Research Awards (MLRA) and the use of the University of Oxford Advanced Research Computing (ARC).

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

## A APPENDIX

### A.1 IMPLEMENTATION DETAILS

**Learning Details** Unless otherwise noted, we train all our self-supervised models with SGD and intial learning rate 0.05 for 400 epochs with two learning rate drops where we divide the rate by ten at 150 and 300 and 350 epochs. We spread our pseudo-label optimizations throughout the whole training process in a logarithmic distribution. We optimize the label assignment at $t_i = \left( \frac{i}{M-1} \right)^2, i \in \{1, \ldots, M\}$, where $M$ is the user-defined number of optimizations and $t_i$ is expressed as a fraction of total training epochs. For the Sinkhorn-Knopp optimization we set $\lambda = 25$ as in (Cuturi, 2013). We use standard data augmentations during training that consist of randomly resized crops, horizontal flipping and adding noise, as in (Wu et al., 2018).

**Quantitative Evaluation – Technical Details.** Unfortunately, prior work has used several slightly different setups, so that comparing results between different publications must be done with caution.

In our ImageNet implementation, we follow the original proposal (Zhang et al., 2017) in pooling each representation to a vector with 9600, 9216, 9600, 9600, 9216 dimensions for `conv1-5` using adaptive max-pooling, and absorb the batch normalization weights into the preceding convolutions. For evaluation on ImageNet we follow RotNet to train linear probes: images are resized such that the shorter edge has a length of 256 pixels, random crops of $224 \times 224$ are computed and flipped horizontally with 50% probability. Learning lasts for 36 epochs and the learning rate schedule starts from 0.01 and is divided by five at epochs 5, 15 and 25. The top-1 accuracy of the linear classifier is then measured on the ImageNet validation subset by optionally extracting 10 crops for each validation image (four at the corners and one at the center along with their horizontal flips) and averaging the prediction scores before the accuracy is computed or just taking the a centred crop. For CIFAR-10/100 and SVHN we train AlexNet architectures on the resized images with batchsize 128, learning rate 0.03 and also the same image augmentations (random resized crops, color jitter and random grayscale) as is used in prior work (Huang et al., 2019). We use the same linear probing protocol as for our ImageNet experiments but without using 10 crops. For the weighted kNN experiments we use $k = 50, \sigma = 0.1$ and we use an embedding of size 128 as done in previous works.

In Table 9, when retraining an AlexNet using ResNet generated labels, we can apply heavier augmentation strategies as the labels are kept constant. Hence for the experiments denoted by "+ more aug.", in addition to the usual augmentations, we further randomly apply one of equalize, autoconstrast and sharpening. We find that this raises the performance for ImageNet but lowers the performance on Places by a small amount, hence illuminating the need to always also report performance on both datasets.

### A.2 FURTHER DETAILS

**NMI over time** In fig. A.1 we find that most learning takes place in the early epochs, and we reach a final NMI value of around 66%. Similarly, we find that due to the updating of the pseudo-labels at regular intervals and our data augmentation, the pseudo-label accuracies keep continuously rising without overfitting to these labels.

**Clustering metrics** In table A.1, we report standard clustering metrics (see (Vinh et al., 2010) for detailed definitions) of our trained models with regards to the ImageNet validation set ground-truth labels. These metrics include chance-corrected metrics which are the adjusted normalized mutual information (NMI) and the adjusted Rand-Index, as well as the default NMI, also reported in DeepCluster (Caron et al., 2018).

**Conv1 filters** In fig. A.3 we show the first convolutional filters of two of our trained models. We can find the typical Gabor-like edge detectors as well as color blops and dot-detectors.

**Entropy over time** In fig. A.4, we show how the distribution of entropy with regards to the true ImageNet labels changes with training time. We find that while at first, all 3000 pseudo-labels contain random real ImageNet labels, yielding high entropy of around $6 \approx \ln(400) = \ln(1.2 \cdot 10^6 / 3000)$. Towards the end of training we arrive at a broad spectrum of entropies with some as low as

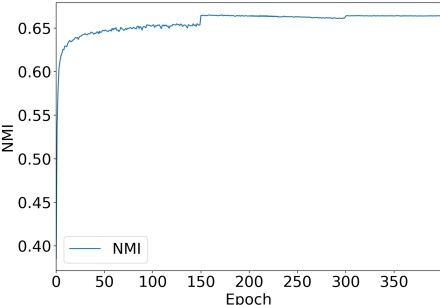 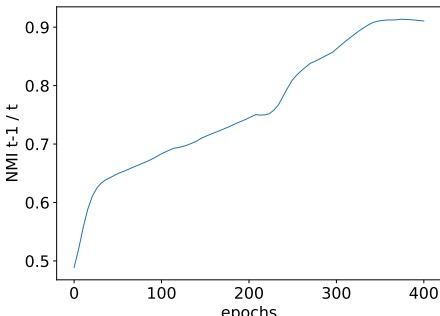

Figure A.1: **Left**: Normalized Mutual Information (NMI) against validation set ImageNet labels. This measure is not used for training but indicates how good a clustering is. **Right**: Similarities of consecutive labellings using NMI. Both plots use the $[10\text{k} \times 1]$ AlexNet for comparability with the DeepCluster paper (Caron et al., 2018).

Table A.1: Clustering metrics that compare with ground-truth labels of the ImageNet validation set (with 1-crop). For reference, we provide the best Top-1 error on ImageNet linear probing (as reported in the main part).*: for the multi-head variants, we simply use predictions of a randomly picked, single head.

| | | Metrics | | |
| | | | adjusted | |
| Variant | NMI | adjusted NMI | Rand-Index | Top-1 Acc. |
|---|---|---|---|---|
| SeLa $[1\text{k} \times 1]$ AlexNet | 50.5% | 12.2% | 2.7% | 42.1% |
| SeLa $[3\text{k} \times 1]$ AlexNet | 59.1% | 9.8% | 2.5% | 44.7% |
| SeLa $[5\text{k} \times 1]$ AlexNet | 66.2% | 7.4% | 1.8% | 43.9% |
| SeLa $[10\text{k} \times 1]$ AlexNet | 66.4% | 4.7% | 1.0% | 43.8% |
| SeLa $[3\text{k} \times 1]$ ResNet-50 | 60.0% | 13.5% | 3.8% | 51.8% |
| SeLa $[3\text{k} \times 10]^*$ ResNet-50 | 66.3% | 26.4% | 10.3% | 61.5% |

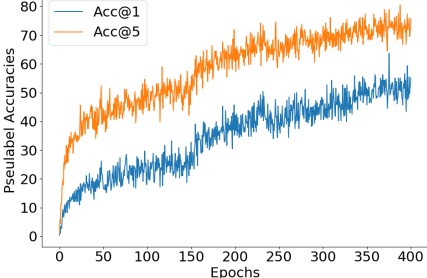

Figure A.2: Pseudo-label accuracies for the training data versus training time for the $[10\text{k} \times 1]$ AlexNet.

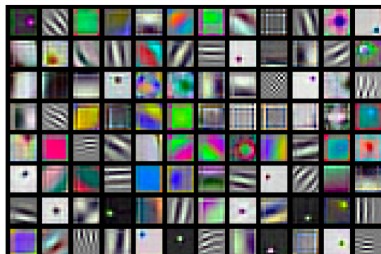 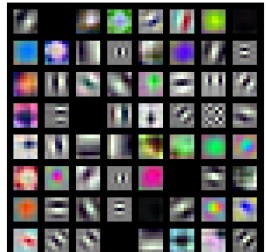

Figure A.3: Visualization of the first convolutional layers of our $[3\text{k} \times 10]$ AlexNet (left) and the $[1\text{k} \times 1]$ ResNet-50 (right). The filters are scaled to lie between (0,1) for visualization.

$0.07 \approx \ln(1.07)$ (see Fig. A.5 and A.6 for low entropy label visualizations) and the mean around $4.2 \approx \ln(66)$ (see Fig. A.7 and A.8 for randomly chosen labels' visualizations).

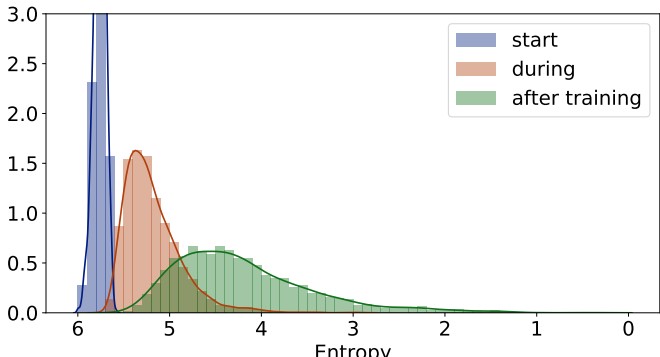

Figure A.4: Cross-entropy of the pseudo-labels with the true ImageNet training set labels. This measure is not used for training but indicates how good a clustering is. This plot uses the $[10k \times 1]$ AlexNet to compare to the equivalent plot in (Caron et al., 2018).

## A.3 COMPLETE TABLES

In the following, we report the unabridged tables with all related work.

Table A.2: **Linear probing evaluation – AlexNet.** A linear classifier is trained on the (downsampled) activations of each layer in the pretrained model. We **bold** the best result in each layer and *underline* the second best. The best layer is highlighted in blue. * denotes a larger AlexNet variant. − refers to AlexNets trained with self-label transfer from a corresponding ResNet-50. "+Rot" refers to retraining using labels and an additional RotNet loss, "+ more aug." includes further augmentation during retraining.

| | | ILSVRC-12 | | | | | Places | | | | |
|---|---|---|---|---|---|---|---|---|---|---|---|
| | Method | c1 | c2 | c3 | c4 | c5 | c1 | c2 | c3 | c4 | c5 |
| | ImageNet supervised, (Zhang et al., 2017) | 19.3 | 36.3 | 44.2 | 48.3 | 50.5 | 22.7 | 34.8 | 38.4 | 39.4 | 38.7 |
| | Places supervised, (Zhang et al., 2017) | - | - | - | - | - | 22.1 | 35.1 | 40.2 | 43.3 | 44.6 |
| | Random, (Zhang et al., 2017) | 11.6 | 17.1 | 16.9 | 16.3 | 14.1 | 15.7 | 20.3 | 19.8 | 19.1 | 17.5 |
| | Random* | 15.6 | 16.8 | 17.4 | 15.6 | 10.6 | 16.5 | 17.6 | 18.6 | 18.1 | 16.3 |
| *1-crop evaluation* | Inpainting, (Pathak et al., 2016) | 14.1 | 20.7 | 21.0 | 19.8 | 15.5 | 18.2 | 23.2 | 23.4 | 21.9 | 18.4 |
| | BiGAN, (Donahue et al., 2017) | 17.7 | 24.5 | 31.0 | 29.9 | 28.0 | 22.0 | 28.7 | 31.8 | 31.3 | 29.7 |
| | Context*, (Doersch et al., 2015) | 16.2 | 23.3 | 30.2 | 31.7 | 29.6 | 19.7 | 26.7 | 31.9 | 32.7 | 30.9 |
| | Colorization, (Zhang et al., 2016) | 13.1 | 24.8 | 31.0 | 32.6 | 31.8 | 16.0 | 25.7 | 29.6 | 30.3 | 29.7 |
| | Jigsaw, (Noroozi & Favaro, 2016) | 18.2 | 28.8 | 34.0 | 33.9 | 27.1 | 23.0 | 31.9 | 35.0 | 34.2 | 29.3 |
| | Counting, (Noroozi et al., 2017) | 18.0 | 30.6 | 34.3 | 32.5 | 25.7 | 23.3 | 33.9 | 36.3 | 34.7 | 29.6 |
| | SplitBrain, (Zhang et al., 2017) | 17.7 | 29.3 | 35.4 | 35.2 | 32.8 | 21.3 | 30.7 | 34.0 | 34.1 | 32.5 |
| | Instance retrieval, (Wu et al., 2018) | 16.8 | 26.5 | 31.8 | 34.1 | 35.6 | 18.8 | 24.3 | 31.9 | 34.5 | 33.6 |
| | CC+VGG-, (Noroozi et al., 2018) | 19.2 | 32.0 | 37.3 | 37.1 | 34.6 | 22.9 | 34.2 | **37.5** | 37.1 | 34.4 |
| | Context 2 (Mundhenk et al., 2018) | 19.6 | 31.8 | 37.6 | 37.8 | 33.7 | 23.7 | 34.2 | 37.2 | 37.2 | 34.9 |
| | RotNet, (Gidaris et al., 2018) | 18.8 | 31.7 | 38.7 | 38.2 | 36.5 | 21.5 | 31.0 | 35.1 | 34.6 | 33.7 |
| | Artifacts, (Jenni & Favaro, 2018) | 19.5 | 33.3 | 37.9 | 38.9 | 34.9 | 23.3 | 34.3 | 36.9 | 37.3 | 34.4 |
| | AND*,(Huang et al., 2019) | 15.6 | 27.0 | 35.9 | 39.7 | 37.9 | - | - | - | - | - |
| | CMC*,(Tian et al., 2019) | 18.4 | 33.5 | 38.1 | 40.4 | 42.6 | - | - | - | - | - |
| | AET*,(Zhang et al., 2019) | 19.3 | **35.4** | **44.0** | 43.6 | 42.4 | 22.1 | 32.9 | 37.1 | 36.2 | 34.7 |
| | RotNet+retrieval*, (Feng et al., 2019) | **20.8** | 35.2 | 41.8 | 44.3 | 44.4 | 24.0 | **33.8** | **37.5** | **39.3** | **38.9** |
| | SeLa [3k × 10]* | 20.3 | 32.2 | 38.6 | 41.4 | 39.6 | **24.5** | 31.9 | 36.7 | 38.0 | 37.0 |
| | SeLa [3k × 10]⁻+Rot* | 20.6 | 32.3 | 40.4 | 43.1 | 42.3 | 24.0 | 31.7 | 37.1 | 39.0 | 37.6 |
| | SeLa [3k × 10]⁻+Rot*+more aug. | 19.2 | 32.6 | 40.8 | **44.4** | **44.7** | 21.1 | 30.4 | 36.5 | 37.9 | 37.3 |
| *10-crop evaluation* | ImageNet supervised* | 21.6 | 37.2 | 46.9 | 52.9 | 54.4 | 22.6 | 33.2 | 39.0 | 41.3 | 39.7 |
| | Random* | 17.6 | 20.3 | 20.6 | 17.8 | 11.0 | 19.2 | 20.7 | 21.8 | 21.3 | 19.0 |
| | DeepCluster (RGB)*, (Caron et al., 2018) | 18.0 | 32.5 | 39.2 | 37.2 | 30.6 | - | - | - | - | - |
| | DeepCluster*, (Caron et al., 2018) | 13.4 | 32.3 | 41.0 | 39.6 | 38.2 | 23.8 | 32.8 | 37.3 | 36.0 | 31.0 |
| | Local Agg.*, (Zhuang et al., 2019) | 18.7 | 32.7 | 38.1 | 42.3 | 42.4 | 18.7 | 32.7 | 38.2 | 40.3 | 39.5 |
| | RotNet+retrieval*, (Feng et al., 2019) | 22.2 | **38.2** | 45.7 | 48.7 | 48.3 | 25.5 | **36.0** | 40.1 | 42.2 | **41.3** |
| | SeLa [3k × 10]* | **22.5** | 37.4 | 44.7 | 47.1 | 44.1 | 26.7 | 34.9 | 39.9 | 41.8 | 39.7 |
| | SeLa [3k × 10]⁻+Rot* | 22.8 | 37.8 | **46.7** | 49.7 | 48.4 | **26.8** | 35.5 | **41.0** | **43.0** | **41.3** |
| | SeLa [3k × 10]⁻+Rot*+more aug. | 21.9 | 37.1 | 46.0 | **50.0** | **50.0** | 23.4 | 33.0 | 39.4 | 41.4 | 39.9 |

Table A.3: **Linear evaluation - ResNet.** A linear layer is trained on top of the global average pooled features of ResNets. All evaluations use a single centred crop. We have separated much larger architectures such as RevNet-50×4 and ResNet-161. Methods in brackets use a augmentation policy learned from supervised training and methods with * are not explicit about which further augmentations they use.

| Method | Architecture | Top-1 | Top-5 |
|---|---|---|---|
| Supervised, (Donahue & Simonyan, 2019) | ResNet-50 | 76.3 | 93.1 |
| Supervised, (Donahue & Simonyan, 2019) | ResNet-101 | 77.8 | 93.8 |
| Jigsaw, (Kolesnikov et al., 2019) | ResNet-50 | 38.4 | — |
| RelPathLoc, (Kolesnikov et al., 2019) | ResNet-50 | 42.2 | — |
| Exemplar, (Kolesnikov et al., 2019) | ResNet-50 | 43.0 | — |
| Rotation, (Kolesnikov et al., 2019) | ResNet-50 | 43.8 | — |
| Multi-task, (Doersch & Zisserman, 2017) | ResNet-101 | — | 69.3 |
| CPC, (Oord et al., 2018) | ResNet-101 | 48.7 | 73.6 |
| BigBiGAN, (Donahue & Simonyan, 2019) | ResNet-50 | 55.4 | 77.4 |
| LocalAggregation, (Zhuang et al., 2019) | ResNet-50 | 60.2 | — |
| Efficient CPC v2.1, (Hénaff et al., 2019) | ResNet-50 | (63.8) | (85.3) |
| CMC, (Tian et al., 2019) | ResNet-50 | (64.1) | (85.4) |
| MoCo, (He et al., 2019) | ResNet-50 | 60.6 | — |
| PIRL, (Misra & van der Maaten, 2019)* | ResNet-50 | **63.6** | — |
| SeLa [3k × 10] | ResNet-50 | 61.5 | **84.0** |
| *other architectures* | | | |
| Rotation, (Kolesnikov et al., 2019) | RevNet-50×4 | 53.7 | — |
| BigBiGAN, (Donahue & Simonyan, 2019) | RevNet-50×4 | 60.8 | 81.4 |
| AMDIM, (Bachman et al., 2019) | Custom-103 | (67.4) | (81.8) |
| CMC, (Tian et al., 2019) | RevNet-50×4 | 68.4 | 88.2 |
| MoCo, (He et al., 2019) | RevNet-50×4 | 68.6 | — |
| Efficient CPC v2.1, (Hénaff et al., 2019) | ResNet-161 | **71.5** | **90.1** |

## A.4 Low Entropy Pseudoclasses

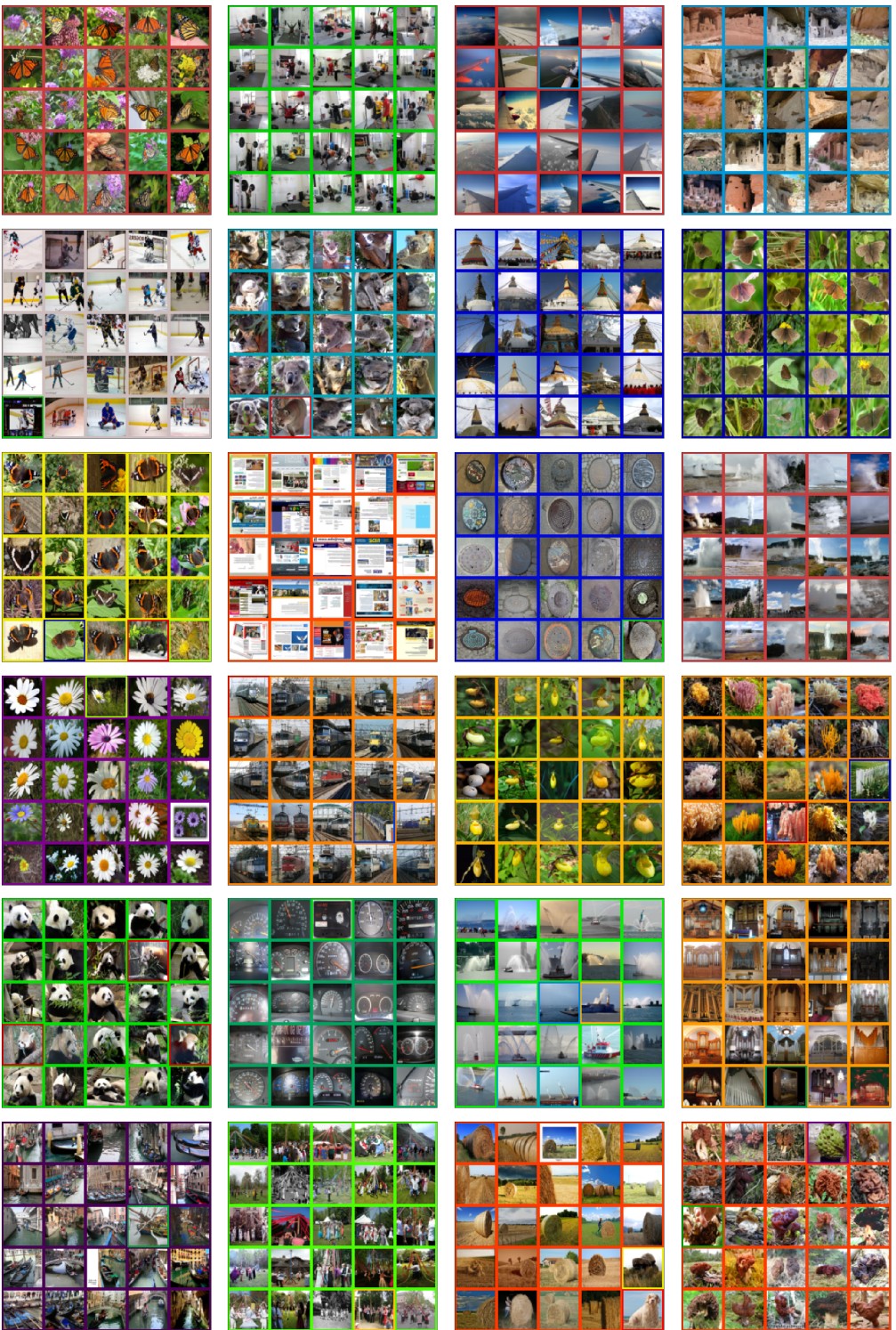

Figure A.5: Here we show a random sample of images associated to the lowest entropy pseudoclasses. The entropy is given by true image labels which are also shown as a frame around each picture with a random color. This visualization uses ResNet-50 $[3k \times 1]$. The entropy varies from $0.07 - -0.83$

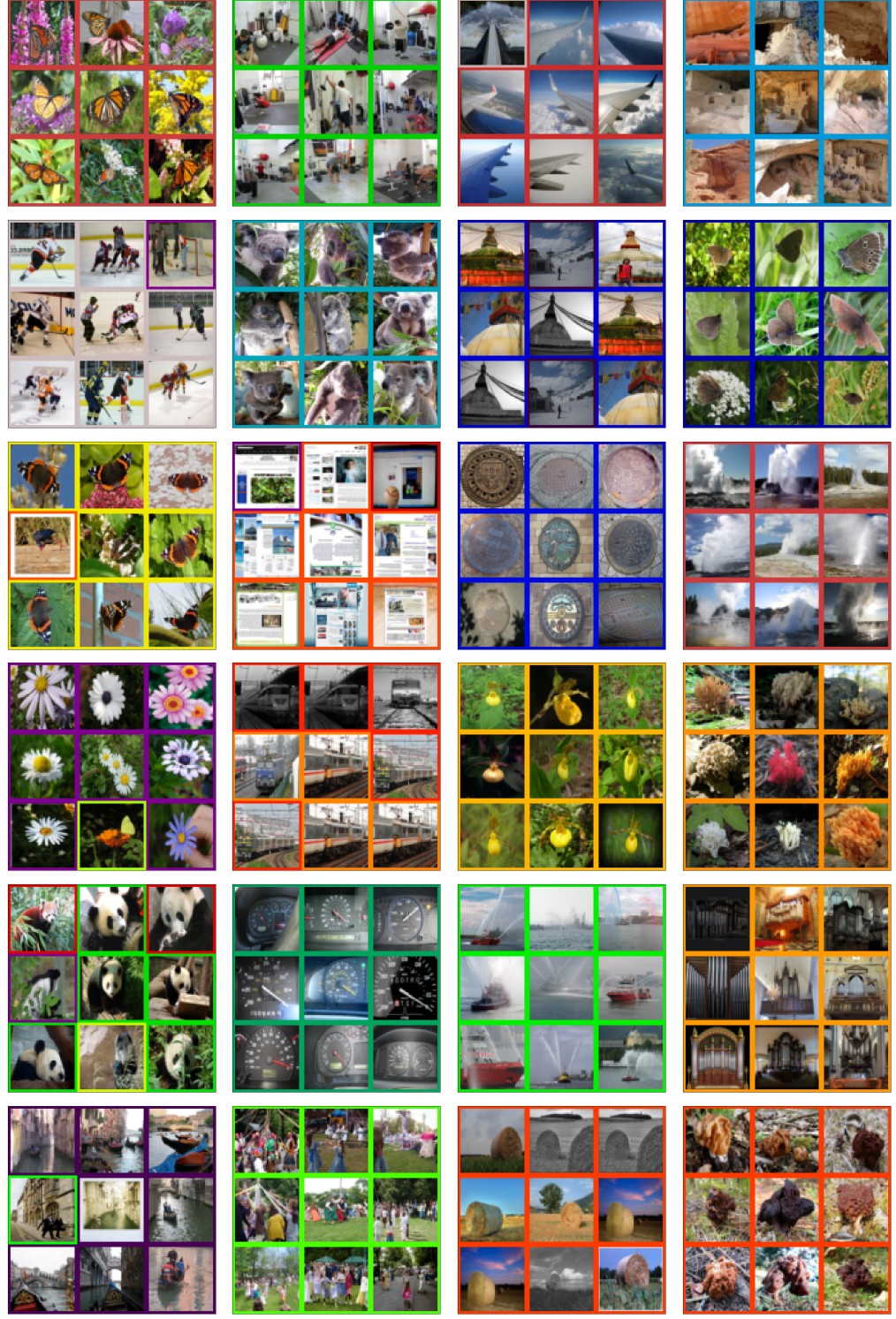

Figure A.6: Visualization of pseudoclasses on the validation set. Here we show random samples of validation set images associated to the lowest entropy pseudoclasses of training set. For further details, see Figure A.5. Classes with less than 9 images are sampled with repetition.

## A.5  RANDOM PSEUDOCLASSES

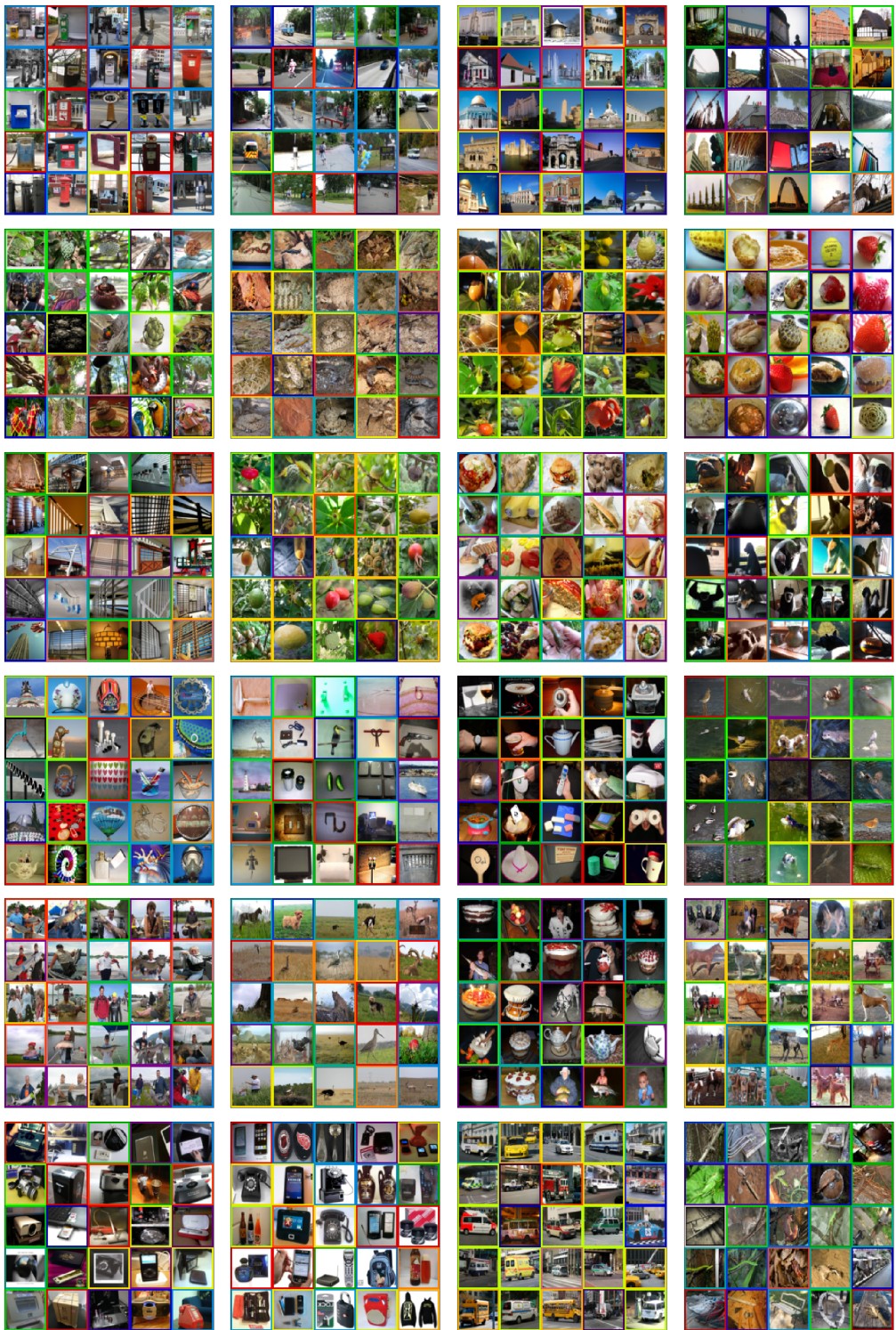

Figure A.7: Here we show a random sample of Imagenet training set images associated to the random pseudoclasses. The entropy is given by true image labels which are also shown as a frame around each picture with a random color. This visualization uses ResNet-50 [$3k \times 1$].

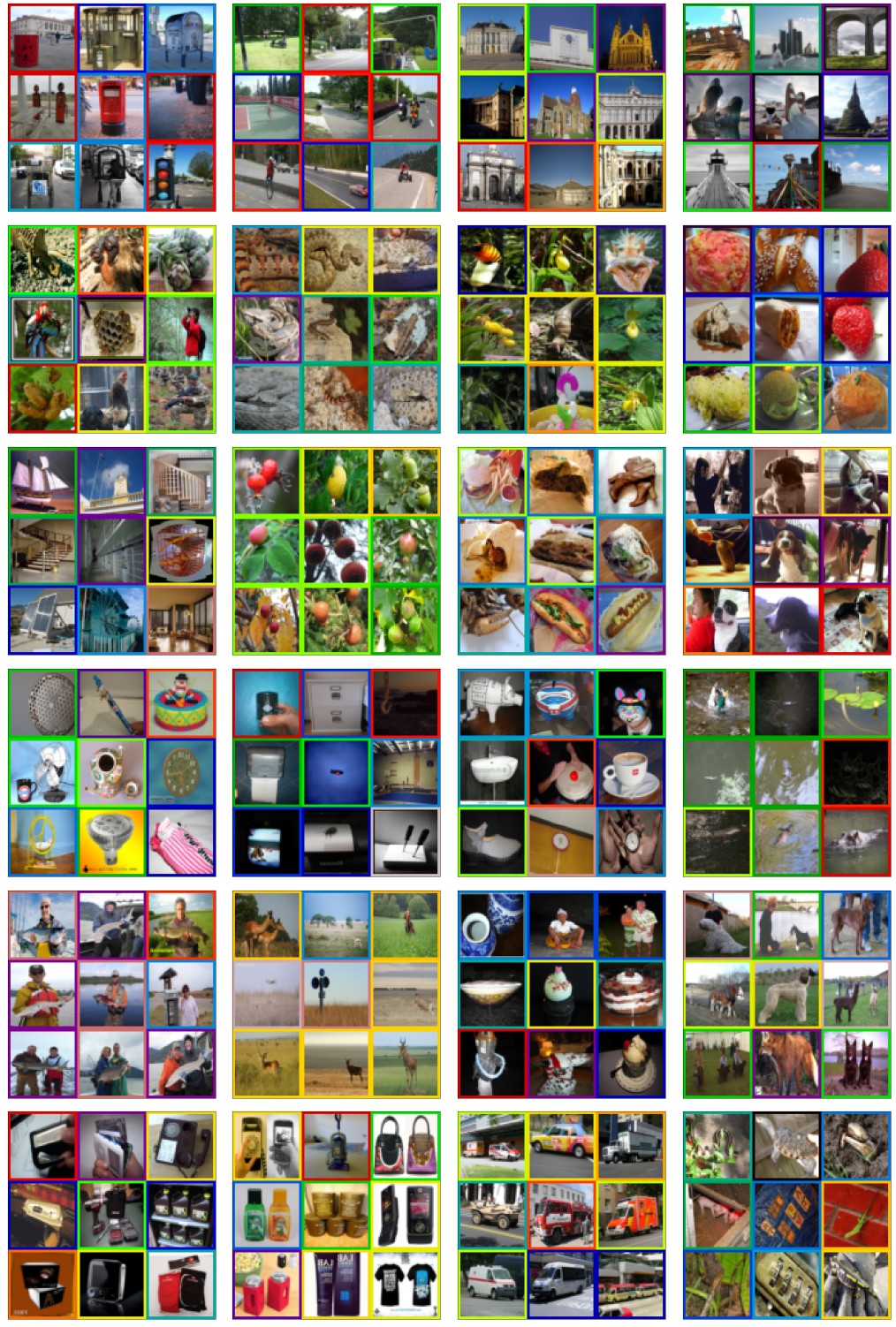

Figure A.8: Here we show a random sample of valdation set images associated to random pseudo-classes. The entropy is given by true image labels which are also shown as a frame around each picture with a random color. This visualization uses ResNet-50 $[3k \times 1]$. Classes with less than 9 images are sampled with repetition.

