# OpenReview forum: "Self-labelling via simultaneous clustering and representation learning"
_ICLR.cc/2020/Conference — Accept (Spotlight)_

### Official Review · AnonReviewer3 · 2019-10-20
**Official Blind Review #3**

**Rating:** 8

**Review:**

Summary & Pros
- This paper proposes a representation learning method based on clustering. The proposed method performs clustering and representation learning alternatively and simultaneously. This approach requires only a few domain-specific prior (precisely, CNN prior) while self-supervised learning requires more prior domain knowledge.
- Compared to the previous work, DeepCluster, this paper uses the same objective for clustering and representation learning. For clustering, the objective can be formulated as an optimal transport problem and it can be efficiently solved. This approach provides desired properties such as convergence.
- This paper shows the proposed method outperforms DeepCluster and it achieves comparable performance with SOTA methods in the representation learning literature.

Concerns #1: More analysis should be provided.
- The author claimed that the proposed method has better convergence properties than DeepCluster. To verify that, more experimental or theoretical supports should be provided. For example, the convergence rate might be checked as Figure 2(b) in DeepCluster paper using NMI against the previous iteration.
- If the number of each class is imbalanced, the equipartition constraint might degrade the quality of the label assignment. Thus, ablation studies about the imbalance setting on small-sized datasets such as CIFAR should be provided. I think K-means could prevent such imbalance issues, so in this case, DeepCluster might perform well.

Concerns #2: Performance is still far from SOTA.
- As reported in Section 4.3, the proposed method still underperforms SOTA methods significantly. The SOTA method can be considered as a combination of (instance-wise) clustering and self-supervision. Thus, such a combination should be tried for improving performance.

Concerns #3: How to guarantee this approach finds good semantic representations?
- In this approach, the model generates a task via clustering, so it might suffer from unsuitable solutions even under the equipartition constraint. If we use a very deeper architecture and a larger size of embedding, then the main optimization problem (3) might be solved before correct label assignments. Moreover, at the first iteration, the labels might be totally random, and then clustering quality is also zero. How to guarantee the clustering quality is gradually improved while training?

The proposed method provides meaningful gain compared to the previous work, DeepCluster. I think this direction against self-supervised learning is important because it requires relatively smaller domain knowledge. However,  I'm not sure how the proposed method can converge stably and efficiently. So I think it would be better if more analysis about the convergence is given in a rebuttal.


**Experience Assessment:**

I have read many papers in this area.

**Review Assessment: Checking Correctness Of Derivations And Theory:**

I assessed the sensibility of the derivations and theory.

**Review Assessment: Checking Correctness Of Experiments:**

I assessed the sensibility of the experiments.

**Review Assessment: Thoroughness In Paper Reading:**

I read the paper at least twice and used my best judgement in assessing the paper.

---

> ### Author Response · Authors · 2019-11-06
> **Response to Reviewer 3**
>
> We thank the reviewer for their time and their detailed comments. We provide comments in the same order.
>
> 1a. In our updated paper we provide the suggested plot with the NMI against the previous iteration, where we observe that indeed our method reaches higher values on this measure (NMI against previous of up to 90%) whilst using significantly less cluster optimizations (80 vs 400). This is because DeepCluster uses k-mean it is forced to discard and re-initialize the last layer every epoch, as the cluster-IDs change. We keep stable clusters, and thus do not use reinitializations that slow learning. In Figure A.1 we also see that using a comparable setting (10k clusters), our method reaches an NMI with the ImageNet validation set of >60% in the first ten epochs, which DeepCluster does not even reach after full training. We further confirm this fast training by having conducted one experiment where we linearly probe our “default” AlexNet [3k x 1] at the 200 epoch mark, (i.e. half-way) and we find that the Top-1 convolutional layer is 43.7%, i.e. only 1% point short of its final performance, indicating fast convergence.
>
> 1b. The reviewer raises the important point about the equipartitioning assumption. However, as we show in Section 3.2, this is less of a constraint and more of a regularizing factor, pushing the network to maximize information between image index and pseudo-label. Furthermore, we use a relatively large number of clusters --- this “overclustering” likely allows the method to decompose large clusters until the individual subclusters have a similar mass overall. Still, we would like to do our best to fully answer this question so we are currently running class imbalance experiments; we hope that we will be able to update the paper with this additional experiments in the following days.
>
> 2. Please note that, compared to the AlexNet SOTA (RotNet+Retrieval, Feng 2019) in the updated paper's Table 8, conv4 is the best layer for both methods and the performance difference is less than 1%. We think it is fair to say that we are relatively close (while still training on a single task).
>
> However, the reviewer’s suggestion is definitely valid and we would like to experiment with a hybrid method too. We are now running this experiment and hope to be able to update the table in the next few days.
>
> Furthermore, in the current version of the paper, as R1 suggested, we include a new set of experiments on CIFAR-10/100, SVHN where we significantly outperform the SOTA.
>
> 3. We thank the reviewer for raising an interesting point. We can respond to this theoretically and empirically.
>
> Theoretically, a main motivation for our work is to base the alternate optimization on a single energy function. This, at least compared to the original DeepCluster, guarantees that the method monotonically optimizes an energy, iteration after iteration, and thus the clustering should gradually improve (at least based on the energy).
>
> The relative frequency of the two optimisation steps does not affect the argument above. However, it might affect the quality of the solution since the optimization is still not globally-optimal. This can only be answered empirically.
>
> In order to do so, we have trained a model without label optimizations (i.e. optimizing for fixed, random labels), and only achieve a Top-1 performance of 21.0% for ImageNet linear probing, which is significantly worse and on par with a random network  (see the updated ablation table in the paper --- Table 1). Hence, label optimization is definitely important. Empirically, we found that our method yields good results for 40-160 optimizations spread out over the 400 epochs (also Table 1).
>
> One key insight that we have is that augmentations make learning a random assignment much harder. Memorizing random labels for ImageNet without augmentations is easily doable [2]. However, once we add augmentations, the training loss never reaches 0. This is an indication that the task is hard enough to provide a meaningful signal in each iteration.
>
> [1] A. Kolesnikov et al. "Revisiting Self-Supervised Visual Representation Learning." In CVPR. 2019.
> [2] C. Zhang, et al. "Understanding deep learning requires rethinking generalization." In ICLR 2017

---

### Official Review · AnonReviewer1 · 2019-10-22
**Official Blind Review #1**

**Rating:** 3

**Review:**

This paper develops a novel self-supervised learning method by combing clustering and representation learning together.
Different from other methods, the two tasks are optimized within the same objective function. Under the weak assumption that the number of samples should be similar across different clusters, the authors further develop a modified Sinkhorn-Knopp algorithm to solve the problem. Experiments on real-world image data demonstrate the effectiveness of the developed solution. In general, the whole paper is well written and the developed solution is interesting. However, I have the following comments:

1. I am still confused about the difference between self-supervised learning and clustering when we do not have labeled data. From my point of view, they are actually the same thing. The authors are suggested to provide more explanations about the differences.
2. The used assumption that samples are uniformed distributed across different cluster are too strong in practice. In many real-world scenarios, the size of the cluster often very a lot, I was wondering how the proposed method can tackle this issue.
3. Experiments are only conducted on the image dataset, which is not quite convincing. The authors are suggested to use the datasets that are normally used in the clustering research to further demonstrate the effectiveness of the method.
4. It is quite surprising that conventional clustering evaluation metrics such as Normalized Mutual Information and Adjusted Rand Index are not used in the experiments.
5. How about the time complexity of the developed algorithm? Can it be scaled to large datasets? Further complexity analyzes are suggested.

**Experience Assessment:**

I have published one or two papers in this area.

**Review Assessment: Checking Correctness Of Derivations And Theory:**

I assessed the sensibility of the derivations and theory.

**Review Assessment: Checking Correctness Of Experiments:**

I assessed the sensibility of the experiments.

**Review Assessment: Thoroughness In Paper Reading:**

I read the paper at least twice and used my best judgement in assessing the paper.

---

> ### Author Response · Authors · 2019-11-06
> **Response to Reviewer 1**
>
> We thank the reviewer for their useful comments. Before addressing the specific comments, we would like to emphasize that our goal is not clustering per se, but using clustering as a pretext for self-supervising a deep network. Hence our evaluation assesses primarily this aspect of the method.
>
> We address the comments in the same order:
>
> 1. Self-supervised learning does not need to be done via clustering, e.g. [1] train a CNN by having it predict a relative location of a patch of an image, or [2] train a CNN self-supervisedly by trying to retrieve the image instance. Vice-versa, clustering can be used outside the setting of self-supervised learning for all types of exploratory data analysis.
>
> 2. First, note that our primary objective is to use clustering as a way to self-supervise a CNN feature extractor. For this to work well, our clusters do not need to have a perfect correspondence with “natural” data clusters. Equipartition is better thought of as a form of regularization which maximises information (see also R2’s review), and empirically this outperforms the k-means algorithm that DeepCluster uses for self-supervision.
> Second, we show via ablation that using 3000 clusters for a dataset that “naturally” has 1000 (in the sense that there are 1000 ImageNet classes) is better. This can be seen as overclustering and thus as a way of sidestepping the uniform cluster size regularization. As suggested by R3, we will provide CIFAR-10 experiments on unbalanced datasets to further explore this empirically.
>
> 3. Please note that our main goal is not clustering per-se, but how it can be used to self-supervise image features. In the updated version of the paper, we now provide results for further datasets (CIFAR-10, CIFAR-100 and SVHN, see Table 6) and show that we also achieve state-of-the-art results in self-supervision on these smaller datasets.
>
> 4. We do not stress clustering metrics because our goal is to use clustering as a mean (pre-text) to self-supervision of image features. However, note that we did provide the plot of Normalized Mutual Information (NMI) to the ImageNet validation set against training time in the Appendix (Fig. A.1). We now have included a further table with the NMI, Adjusted NMI and the Adjusted Rand Index of our models in the updated Appendix (see Table A.1).
>
> 5. In our paper we show that our method scales to more than 1M images (ImageNet contains 1.2M training images). The core of our method relies on matrix-vector multiplies the size of $NK^2$, where $N$=number of images and $K$=number of clusters, and so it scales linearly with the number of images. We have added this analysis to the paper.
>
> [1] D. Pathak, et al. “Context Encoders: Feature Learning” by Inpainting. In CVPR 2016.
> [2] Z. Wu, et al. "Unsupervised feature learning via non-parametric instance discrimination." In CVPR 2018.

---

### Official Review · AnonReviewer2 · 2019-10-28
**Official Blind Review #2**

**Rating:** 8

**Review:**

Summary:
The paper proposes a self-supervised learning procedure to train deep neural networks within an unsupervised learning setting. The authors build their work on a pretext task that consists in maximizing the information between the input data samples and labels that are basically obtained by a self-labeling procedure that clusters them in K distinct classes. This is similar to what was done in DeepCluster, a previous self-supervised algorithm that self-labels samples through clustering. However, differently from that approach, the current method does not introduce any additional clustering cost functions. Instead it implicitly achieves self-labeling by simply adding the constraint that label assignments equally partition the dataset. This constraint acts as a "regularizer" that allows the authors to minimize the cross-entropy loss between inputs and pseudo-labels while avoiding the degenerate trivial solution where all samples are assigned to the same pseudo-label. As a result, the authors are able to derive a self-labeling method that optimizes the same cross-entropy loss as the classification task. That is done by remarking that minimizing the loss function over the pseudo-label assignments (under the equal partition constraint) can be formulated as an optimal transport problem that can be solved efficiently with a fast version of the Sinkhorn-Knopp algorithm. In practice, what the authors do at training time is to alternate between 1) minimizing the average cross-entropy loss by training the neural network (feature extractor + linear classification head) given a fixed pseudo-labels assignment, and 2) optimizing the pseudo-label assignments implemented as a matrix Q of posterior distribution of labels given the sample index, which, as said, can be done efficiently with a KL-regularized version of Sinkhorn. Moreover, this last step can be carried out simultaneously for multiple distinct classification heads (with possibly different number of labels), each sharing the same feature extractor but inducing a different matrix Q. At this point, the number of classification heads can be treated as a hyperparameter of the algorithm.
The authors then go on to show that this new algorithm is competitive with current state of the art method with several architectures in terms of providing a good feature extractor for downstream image classification, detection and segmentation tasks. They for instance consistently beat DeepCluster, the main direct competitor, on classification and detection tasks.
They also conduct ablation studies that provide interesting insights on the functioning of their algorithms and the effects of the multiple classification heads, the number of clusters, and the quality of the learned assignment. Intriguingly, on ImageNet they find for example that AlexNet obtains better performance at validation when it's trained from scratch on labels obtained with their self-labeling procedure, as opposed to the original labels.

Decision:
In my opinion this paper should be a clear accept. The paper is well written, presents an elegant idea in a clear and straight-forward manner, and is solidly built on top of the current literature on self-supervised learning for image processing, which is also very well summarized.
The feature extractors obtained with the proposed algorithm are convincingly tested and validated on several downstream tasks (like classification on ImageNet, PascalVOC classification, detection and segmentation), and that is done for several base architectures, obtaining performances that are competitive with state-of-the-art. In addition, a series of careful ablation studies help in gleaning some scientific understanding on the method.

Minor comments:
- The authors refer to traditional clustering like k-means as being "generative", which a little confusing. Clustering algorithms can be derived within a probabilistic framework by positing a generative model of the data, however, strictly speaking, k-means is not by itself is not a generative approach. It's a minor point, but it could be helpful to be more precise about this, in order to avoid possible confusion. The main point that the authors want to make in this regard is that their framework eschews having to posit an additional clustering cost function.

**Experience Assessment:**

I have published one or two papers in this area.

**Review Assessment: Checking Correctness Of Derivations And Theory:**

I carefully checked the derivations and theory.

**Review Assessment: Checking Correctness Of Experiments:**

I assessed the sensibility of the experiments.

**Review Assessment: Thoroughness In Paper Reading:**

I read the paper thoroughly.

---

> ### Author Response · Authors · 2019-11-06
> **Response to Reviewer 2**
>
> We thank the reviewer for their thorough reading and lucid understanding of the key ideas in the paper.
>
> Regarding k-means: We have seen k-means described both as generative (e.g. see the referenced Bishop’s book) and discriminative and k-means can be derived as a special case of GMM (where all variances are isotropic and equal). However, we have revised the manuscript to address this and to avoid confusion with GAN-based generative approaches in self-supervised learning.

---

### Author Response · Authors · 2019-11-06
**Updated version**

We thank the reviewers for their time and their careful analysis of the work presented.
Based on the feedback, we have uploaded an updated version of the paper with the following main changes:
* Experiments on CIFAR-10, CIFAR-100 and SVHN, where we achieve SOTA by a large margin.
* Additional ablations such as [5k x 1] and 0 number of label optimizations.
* Reformatted the ablations into multiple tables for additional ease of understanding.
* Additional plot in the Appendix of NMI vs the previous iteration for the [10k x1] AlexNet.
* Additional table in the Appendix showing NMI, adjusted NMI and adjusted Rand-Index for several models.
* Incorporated several clarifications requested by the reviewers.

---

### Author Response · Authors · 2019-11-15
**Final paper update**

Additional to the previous update, we have revised our paper again with the following changes based on the reviewers’ feedback:
* New ResNet results that use 10 heads (Tab.10), beating the SOTA on a standard ResNet-50 at the time of submission, with Top-1 accuracy of 59.2 as opposed to BigBiGAN’s 55.4.
* New AlexNet results (Tab.9), where we followed R3’s suggestion of combining methods and were able to achieve SOTA on AlexNet, with Top-1 accuracy of 49.6 for ImageNet linear probing, and on Pascal VOC detection with MAP of 59.2.
* Extended the Pascal VOC table (Tab. 7) by one column (classification, fine-tuning only fc6-8), where our method also achieves SOTA.
* Additional imbalance ablations (Sec. 4.4) on CIFAR as requested by reviewers R1 and R3. The results show that our method works well even in heavy imbalance scenarios and the proposed label optimization via Sinkhorn-Knopp significantly outperforms k-means (see below for details).
* Incorporated some minor changes in the text reflecting the new findings.
* Due to the short rebuttal period, for the newly trained models, minor, auxiliary evaluations (Places linear probing) have not yet finished and are marked as “evaluating” in the paper revision. However, all main benchmarks have completed and are reported.

Summarizing, our method achieves state of the art representation learning performance for AlexNet and ResNet-50 on SVHN, CIFAR-10, CIFAR-100 and ImageNet and when transferred to Pascal VOC.


Imbalance experiments
As promised, we have added an analysis on imbalanced data (Sec.4.4). We compare a light and a heavy imbalance scenario against using the full (balanced) CIFAR-10 dataset. The main results are:

We find that in both scenarios our method works well and only suffers small performance losses, inline with the decreased amount of training data. We find that our method consistently outperforms experiments which use k-means (with k-means++ initialization) instead of our Sinkhorn-Knopp based clustering. Furthermore, we find that even in the worst imbalance setting (10% of class 1, 20% of class 2, etc.), our proposed method not only outperforms the experiment which uses k-means on the same data, but also that of k-means with the full data set. This confirms the mathematical clarification provided in Sec. 3.2 that the equipartition is equivalent to maximizing information, regardless of a true label distribution.

---

### Public Comment · ~John_Richard_Corring1 · 2020-01-08
**Edits**

Isn't matrix-vector multiplication O(NK)?

I can't find a resource for the nonsquare Sinkhorn-Knopp algorithm, did you have one?

It seems like the assignments of P to the transport polytope (after Equation 3) may be missing a minus sign... or I am missing something? Correct me if I'm wrong, but you should need this for positivity (for Sinkhorn-Knopp to apply) and for the definition of cross entropy to coincide with the transport cost.

Thanks for a nice paper.

---

> ### Author Response · Authors · 2020-02-26
> **answer**
>
> Hi,
>
> Thank you for the comment. We have updated the notation in the final (camera-version) of the paper where everything should match up properly.
> Thank you for pointing this out.
> Regarding the algorithm, you can find our implementation here: https://github.com/yukimasano/self-label/blob/master/sinkhornknopp.py#L101
>
> Best

---

### Public Comment · ~Amjad_Almahairi1 · 2020-01-22
**Question about solution to Eq. 6**

I would like first to congratulate you for this nice paper. One thing that would be nice to clarify is why we get an integral solution to the linear program in Eq. 6. It doesn't seem that obvious to me, so it would be nice to make a comment on that (or even a proof or reference) in the paper. Also, would you also get an integral solution with regularized version and in practice?

---

### Public Comment · ~Paul_Greene1 · 2020-02-02
**Question about the Q matrix**

I would like to first thank you for this paper. I was wondering if you could answer a couple of questions that I have related to the paper.

1) How do you transition from the probability matrix Q to the labels? Do you just use argmax to assign labels to data points?
2) This might be a silly question but : where exactly do you use the Sinkhorn-Knopp algorithm in the paper? Is it used for initializing the Q matrix (meaning that we initialize Q randomly and then apply the algorithm) ? I searched the whole internet but couldn't find an implementation for the Sinkhorn-Knopp algorithm for non-diagonalizable matrices like the Q matrix (the only condition given in the paper is that "K divides N exactly"). The frequently cited (Cuturi, 2013) paper only talks about square matrices.
3) How do you initialize the α and β scaling vectors? Just randomly? In that case, how do you make sure that the Q matrix remains a probability matrix after "Step 2: self-labelling" is applied?
4) What are the scores in Table 6? Are they accuracies? The current CIFAR-10 supervised SOTA is 99.00% and CIFAR-100 supervised SOTA is 91.70%, but the Table says that they are 91.8% and 71.0% ?
5) Are you planning on releasing the code for the paper?

Thank you very much.

---

> ### Author Response · Authors · 2020-02-26
> **answer**
>
> Hi,
> Thank you for your kind comments. I've also replied to your email but got a bounce back so here I post the reply:
> 1) yes.
> 2) please find the SK algorithm in our github repo: https://github.com/yukimasano/self-label/blob/master/sinkhornknopp.py#L101
> 3) we initialize them by np.ones(N) * 1/N and np.ones(K)*1/K
> 4) they are accuracies using the same architecture implementation and evaluation protocol as the rest in the table. This is to ensure comparability. I’m not sure how the current SOTA in CIFAR is done, but probably with tencrops, more extensive augmentation and maybe a ResNet.
> 5) https://github.com/yukimasano/self-label
>
> Thanks for the wait!

---

> > ### Public Comment · ~Haohang_Xu1 · 2020-07-05
> > **Some questions about transition from the probability matrix Q to the labels.**
> >
> > From your answer to question 1 : 1) How do you transition from the probability matrix Q to the labels? Do you just use argmax to assign labels to data points?
> >
> > Do you mean that in step1(representation learning), the Q of eq.6 is actually a one-hot matrix by applying argmax on probability matrix $Q^*$ (the $Q^*$ here means the direct solution of eq.7 in step2)?
> >
> > If I do not understand correctly, what do you mean about transition from the probability matrix Q to the labels?
> >
> > If I understand correctly, how about use $Q^*$ directly in step1 to compute the cross-entropy loss, but not use $argmax(Q^*)$ ? If a soft label will be better?

---

> > > ### Author Response · Authors · 2020-07-07
> > > **Answer**
> > >
> > > Re the questions: please see here, where we have discussed this: https://github.com/yukimasano/self-label/issues/7

---

### Public Comment · ~Haohang_Xu1 · 2020-07-05
**Question about Eq. 5**

I think Eq.5 should be writen as : log E(p,q) + log N = log<Q, -log P>.  Can you give some points about how Eq.5 holds on?

Thank you very much.

---

> ### Author Response · Authors · 2020-07-07
> **Answer**
>
> For those wondering: We've discussed it here: https://github.com/yukimasano/self-label/issues/7

---

### Decision · Program_Chairs · 2019-12-19

**Decision:**

Accept (Spotlight)

**Comment:**

The paper focuses on supervised and self-supervised learning. The originality is to formulate the self-supervised criterion in terms of optimal transport, where the trained representation is required to induce $K$ equidistributed clusters. The formulation is well founded; in practice, the approach proceeds by alternatively optimizing the cross-entropy loss (SGD) and the pseudo-loss, through a fast version of the Sinkhorn-Knopp algorithm, and scales up to million of samples and thousands of classes.

Some concerns about the robustness w.r.t. imbalanced classes, the ability to deliver SOTA supervised performances, the computational complexity have been answered by the rebuttal and handled through new experiments. The convergence toward a local minimum is shown; however, increasing the number of pseudo-label optimization rounds might degrade the results.

Overall, I recommend to accept the paper as an oral presentation. A more fancy title would do a better justice to this very nice paper ("Self-labelling learning via optimal transport" ?).